# Tracking diachronic sentiment change of economic terms in times of crisis: Connotative fluctuations of 'inflation' in the news discourse

**Javier Fernández-Cruz**\*, **Antonio Moreno-Ortiz**

Facultad de Filosofía y Letras, Universidad de Málaga, Málaga, Spain

\* fernandezcruz@uma.es

## Abstract

The present study focuses on the fluctuation of sentiment in economic terminology to observe semantic changes in restricted diachrony. Our study examines the evolution of the target term 'inflation' in the business section of quality news and the impact of the Great Recession. This is carried out through the application of quantitative and qualitative methods: Sentiment Analysis, Usage Fluctuation Analysis, Corpus Linguistics, and Discourse Analysis. From the diachronic Great Recession News Corpus that covers the 2007–2015 period, we extracted sentences containing the term 'inflation'. Several facts are evidenced: (i) terms become event words given the increase in their frequency of use due to the unfolding of relevant crisis events, and (ii) there are statistically significant culturally motivated changes in the form of emergent collocations with sentiment-laden words with a lower level of domain-specificity.

**Data Availability Statement:** All 14 files are available from OSF: Moreno-Ortiz A, Fernandez-

## 1. Introduction

The multiple aspects that define specialized languages in use have been researched from diverse perspectives and approaches: terminology, expertise, discourse analysis, diachrony, etc. This research aims to describe how the sentiment of certain specialized terms that correspond to major macroeconomic indicators, such as 'inflation', may undergo important connotation shifts during major economic events, as reflected in their use in mass media. Since the advent of the modern press in the 19th century, journalists have shaped how we perceive the world, and served as de facto language change agents. Renouf and colleagues' research on newspaper language [1, 2] discussed how news discourse (e.g., evaluative language, metaphors, and collocations) reflect and shape social attitudes and values. In fact, the communicative role of journalists consists in putting complex realities into words, which means "translating" matters concerning important events, such as economic crises, and therefore determine to some extent the public's understanding and emotional reactions. As Orts Llopis [3] states, "economy has evolved from being a subject only for connoisseurs to becoming a topic of general interest, as the evolution of global finances is warily monitored by experts and followed by laypeople through traditional media and the social networks" (p. 218). The language used in news

Cruz J. "inflation" dataset. 2023. doi:10.17605/OSF.IO/Z4QMV.

**Funding:** *JFC *(No grant number) *JFC is a researcher under the Margarita Salas research program from the Spanish Ministry of Universities, funded by the European Union-Next Generation EU. * https://www.uma.es/servicio-de-investigacion/info/129684/ayudas-para-la-recualificacion-del-sistema-universitario-espanol/ *AMO *PID2020-115310RB-I00 *Spanish Ministry of Science and Innovation * https://www.aei.gob.es/convocatorias/buscador-convocatorias/proyectos-idi-2020-modalidades-retos-investigacion-generacion The funders had no role in study design, data collection and analysis, decision to publish, or preparation of the manuscript.

**Competing interests:** The authors have declared that no competing interests exist.

transmission serves as a powerful tool for subtly projecting the sentiment of economic policies, whose potential we are only beginning to comprehend.

Moirand [4, 5] discusses the coverage of special events in newspapers, emphasizing the use of what she calls 'event words' (*mots événements*), such as "Watergate", "mini-budget", or "credit crunch", which represent real-life events that have garnered significant media attention and eventually become the names of specific events ('discursive moments'). Frequently, they can be paraphrased as "the affair of [noun phrase]" or "the [noun phrase] affair" (e.g., "the Watergate affair") in reference to the original event; that is, the noun or noun phrase in question crystalizes and acquires the full meaning of the event that it originally referred to. Event words are often found on the front page of newspapers, in headlines that contain the emotional theme of the discourse. In our case, we consider that specialized terms in economy and finance news that become event words transcend their technical definition and are used more frequently by the general public. Also, at least during a certain amount of time, event words may also reflect the connotations that accompany the term in the news discourse.

Thus, the public's understanding of the economy and its emotional triggers is based not only on our immediate reality, but also on the language used to talk about it, including media depictions. Taking a financial power-press-public chain of linguistic transmission as a point of departure, Haase [6] argues that the numbers do not speak for themselves, especially in the narration of economic crisis events. Therefore, economic analysts are highly reliant on a technolect-fueled rhetoric rich in metaphors that is frequently replicated by journalists.

Kovecses [7], Charteris-Black [8], and Charteris-Black & Ennis [9] proved that the intensification of the emotional impact in economic language and metaphorical images is linked to the rhetorical concept of pathos. Regarding the speed of changes in the narration of crisis events, we hypothesize that the use of terms and their lexical co-selections in the press transcends the mere factual description of macroeconomic data. On many occasions, sentiment is prevalent, as the mere mention of certain terms (e.g., 'inflation' or 'debt') at specific times radiates emotion. Indeed, there is a linguistic challenge in working with such rapid changes in the sentiment of words that cannot be reflected in the definitions of standard dictionaries. In the field of Economics, Doms & Morin [10], executives at the Federal Reserve, have developed indexes that utilize news-based keywords and phrases to gauge the tone and volume of economic reporting. This use of keywords encourages readers/consumers to adjust their economic expectations in light of the new perceived state of the economy.

This study deals with semantic change, specifically the phenomenon of short-term 'amelioration' (i.e., gaining positive sentiment) and 'pejoration' (shifting towards negative sentiment) in financial terms. We have selected the term 'inflation' as our case study, acknowledging its significance as a crucial economic phenomenon during the time this article was written. In the last quarter of 2022, inflation skyrocketed due to rising energy costs and the ongoing conflict in Ukraine. After mid-year 2021, Google searches for 'inflation' have continued to increase, as public interest in the term rose (Fig 1). Before that, as seen in the Google Trends timeline (Fig 2), the last full historical cycle of 'inflation' being an event word that concerned not only specialists, but also the press and the public, was during the Great Recession. Along with the Crash of 1929 and the 1970s Great Inflation, the Credit Crunch of 2007–2008 and the recession that followed are key areas of study for economists and decision-makers. The events and policies that emerged from them are still widely debated issues. The effort to study these factors is an important opportunity for agents to draw lessons that can influence policies. Our conviction is that prioritizing a retrospective examination of established historical events [11] over a pressing body of contemporary issues presents a compelling linguistic landscape and furnishes foundational comprehension for interpreting current events of historical significance.

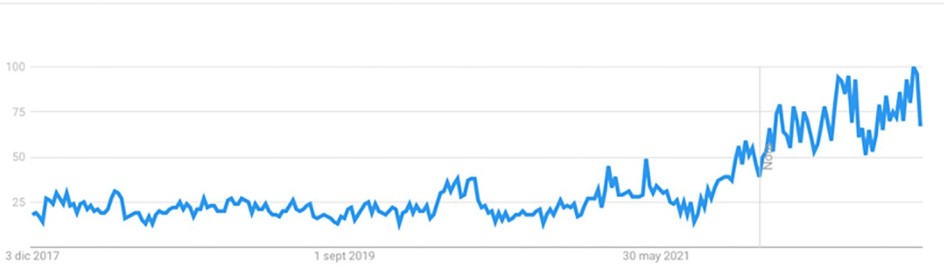

**Fig 1. Interest in 'inflation' as a search term (2017–2021).** Extracted from Google Trends.

Inflation served as a prominent macroeconomic indicator during the global crisis, especially in the European scenario. In contrast to the Federal Reserve in the USA, the obligation of the European Central Bank (ECB) is simply to control inflation, as it does not have a direct mandate to specific EU national economies (e.g., labor policies). ECB member states, on the other hand, are unable to issue currency to pay their debt holders. The press largely covered the inflation problem, which became apparent when the deflation levels were reached in 2009. With rising budget deficits and approaching debt maturities, bonds skyrocketed. During 2010–2014, there was no implementation of policies that would increase inflation and thus stimulate consumption, as low inflation was beneficial to many northern EU countries. On the other hand, southern EU countries, paradigmatically Greece, could not implement measures to adjust their economies and alleviate the crisis that brought into question the sustainability of the entire Eurozone. Inflation figures were very low and the ECB had to implement strategies for the stimulus to avoid crashing consumption levels owing to possible deflation [12].

The laying of the groundwork for stimulus policies was discursively performed on July 26, 2012, by ECB President Mario Draghi, who signaled that the ECB would do "whatever it takes" to save the euro from any speculation processes: "But there is another message I want to tell you. Within our mandate, the ECB is ready to do whatever it takes to preserve the euro. And believe me, it will be enough" [13]. These words echoed in the press and quickly influenced the economy; within a few days, the markets reversed the negative trend, and the following months actually saw the crisis calm and interest rates fall. Eurozone inflation levels reached a joint record low in 2015. At this final stage of the Euro crisis, prices tumbled and deflation challenged the control of sovereign debt for local governments.

We speculate that the usage of the term 'inflation' in the press seems to have changed in the narratives of this timeline. However, variations in sentiment, or semantic orientation, are undoubtedly difficult to trace through introspection. Instead, we propose that the phenomenon of diachronic fluctuation of sentiment in economic terms can be observed in restricted diachrony (2007–2015) using language data. In addition, our main objective concerns the methodological aspects. We propose a method that we consider to be generalizable enough to

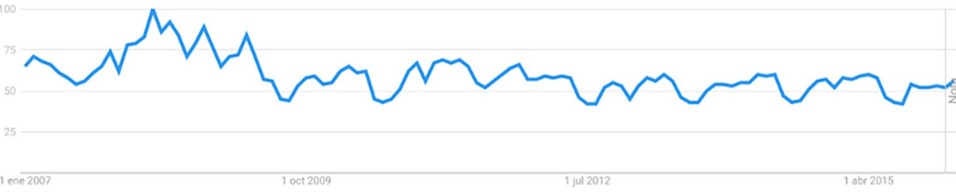

**Fig 2. Interest in 'inflation' as a search term (2007–2015).** Extracted from Google Trends.

help identify the evolution of the semantics of terms in specialized texts through the combination of Sentiment Analysis, Usage Fluctuation Analysis and Corpus Linguistics. To our knowledge, no studies have integrated these techniques to investigate the evolution of language usage in news media. Our research addresses three major issues: (i) the public dissemination of specialized terms by news outlets, (2) the discursive emotional subtext of in vivo terminology, and (iii) short-term semantic change.

## 2. Theoretical background

### 2.1. Contemporary language change

The development of interest in language change in living languages is not exclusively a contemporary phenomenon, as its roots can be traced back to the 19th century (e.g., the Neogrammarians or the second Wittgenstein). Regarding lexical semantics, the study of word meaning over time is usually referred to as a 'semantic shift' or 'semantic change'.

The theoretical background of semantic change, originating from cognitive and psychological mechanisms, can be found in the pre-structuralist postulates of Bréal [14] who enunciated six types of semasiological change: 'pejoration' versus 'amelioration,' 'restriction' versus 'expansion', 'metaphor,' and 'metonymy'. In the first axis, the meaning of a word changes to become more positive or negative. Geeraerts [15] points out several driving forces of semantic change. Largely, it is (i) subject to non-referential meaning and (ii) culturally motivated (i.e., it changes with the cultural evolution of views). Semantic change also entails a psychological process of subjectivation. Provencio Garrigós [16] evidences a process of subjectivization, from the objective meaning of a word in the voice of a speaker (depending on external factors) to a subjective meaning that describes an internal or subjective situation (subject to the speaker's cognos).

Motivations for semantic change are of two types: 'linguistic drifts' (changes in the central meaning of words, mainly slow and regular) and 'cultural shifts' (variations in word associations, mainly culturally motivated) [17]. Half a century ago, Waldron [18] identified changes in meaning based on variations between lexical relations: "Change of meaning, then, essentially concerns a shift of relations between words on the one hand and integral senses on the other hand" (p. 139). In relation to this, Traugott [19] reflected on the conditions of semantic change, considering that behind every variation in meaning, there is a chain of causality that, in fact, belongs to extralinguistic causes (i.e., material, social, or psychological conditions). We adhere to Lyons' [20] assertion that even if semantic change is regular and continuous, "languages change more rapidly in certain periods than they do in others" (p. 183).

Finally, the use of computational techniques to study semantic changes has emerged as a distinct research methodology over the last decade. Different linguistic theories and models have been proposed to explain rules and regularities in semantic change, such as Diachronic Prototype Semantics [15] and the Invited Inference Theory of Semantic Change [21].

### 2.2. Corpus-based methodologies to study semantic change

The availability of large corpora and the development of natural language processing have led to data-driven approaches that produced paradigm changes. Many research initiatives have emerged to grapple with tasks that were difficult to approach several decades ago. A state-of-the-art review by Kutuzov et al. [17] compiled dozens of papers on this subject using heterogeneous methodological approaches. Three main research communities are interested in this particular topic: (i) natural language processing/computational linguistics, (ii) information retrieval/computer science, and (iii) political science. Alessi & Partington [22] comprehensively described the tools and methodologies employed in the novel discipline of 'modern

diachronic corpus-assisted language studies'. This study reflects on the consideration that tracking contemporary language has received little attention in linguistics, which has been justified by two main factors: (i) the difficulties in developing instruments and methods to carry out this task, and (ii) the difficulty in observing changes in the short term. Corpus linguistics and natural language processing techniques provide solutions to these problems by providing sophisticated tools to process large amounts of contextualized text, which facilitates the study of the evolution of different text types over short periods of time.

The analysis of collocations is a fundamental aspect of corpus-based studies, particularly when exploring semantic change. This method plays a key role in understanding how words and phrases co-occur, and how they can evolve over time. Williams [23] revealed the crucial role that collocational networks play in the formation of meaning. The acquired connotations of a word's meaning can 'push' its original semantics into a subordinate position. In relation to this, Hoey [24] proposed an expansion of the idea of 'lexical priming', in which a word gradually acquires meaning for each individual speaker as it becomes primed for collocation, colligation, and semantic and pragmatic associations. Repetition is a key factor, as every re-encounter that a speaker may have with a particular word reinforces this conditioning.

The concept of 'semantic prosody' or 'semantic association' is also central to the qualitative study of this work. Post-Firthian scholars Sinclair [25] and Louw [26] were the first to use the term, which refers to expressions associated mostly with sentiment-related meanings, whether positive or negative. Louw defined semantic prosody as "a consistent aura of meaning with which a form is imbued by its collocates" (p. 157), in a way that maintains evaluative harmony and listener comprehensibility. Semantic prosody is thus a non-static property of the lexicon but variable in time, which researchers can explore, explain, and predict using a text corpus. According to Bublitz [27], a set of collocations evolves to acquire a tinge of sentiment (positive or negative) over a time interval and context analysis of the term. To illustrate this, from a qualitative Political Sciences perspective, Stratigaki [28] argues that economic actors choose original terms in order to fit them into their political agenda.

Finally, we argue that terms can develop semantic prosody as they gain prominence in news discourse following a major event, gaining gradual collocation with new non-domain-specific (i.e., general language) words over time. It is important to understand that certain words, such as 'ease' or 'fight', which may be said to carry specific sentiment taken in isolation, may act simply as sentiment shifters that modify the inherent semantic orientation of the terms they collocate with, such as 'inflation' or 'debt'. Thus, these shifters are to be considered general collocates, and are used to describe the economic situation, regardless of the sentiment conveyed in the context.

In this work, however, we conform to the semantic prosody assumption that a higher frequency of sentiment words of a certain polarity in the vicinity of an event word is an indicator that a shift towards that polarity is taking place regarding that word, and therefore changes in its semantic orientation can be tracked by studying the contexts of the focus terms over time.

## 2.3. Sentiment analysis

The wide availability of electronic text on the Internet has led to an increase in research on information extraction, a subfield of natural language processing that seeks to obtain structured data from unstructured text. Sentiment Analysis (SA) [29], also called opinion mining, is the field of natural language processing and textual analysis that studies "people's opinion, sentiments, evaluations, appraisals, attitudes, and emotions towards entities such as products, services, organizations, individuals, issues, events, topics, and their attributes" (p. 451). The basic tasks of a SA system include polarity detection and emotion recognition [30]. A typical SA

system classifies texts according to their polarity (i.e., positive or negative) or output on a scale (e.g., 0 for very negative and 100 for very positive). Discourse, syntactic, textual, and intertextual phenomena play crucial roles in sentiment analysis. Many disciplines, including linguistics, psychology, political science, and economics, make use of their applications.

There are two main approaches to SA: machine learning and lexicon-based. Machine-learning approaches rely on datasets that contain annotated sample sentences and different features that are subsequently processed and learned by a computational system. Unsupervised machine-learning approaches in the form of neural networks (specifically, Transformers-based models, such as BERT) offer current state-of-the-art results for SA tasks [31]. In lexicon-based systems, sentiment words are detected using sentiment dictionaries of varying quality and level of detail. Ideally, context is processed to account for the modification of polarity or valence. These semantic orientation modification mechanisms have been studied and formalized through various systems of rules. Polanyi & Zaenen [32] first proposed at a theoretical level what they called 'sentiment shifters' or 'contextual valence shifters', a series of rules that modify the sentiment of the expression by inverting polarity, its intensification, or its attenuation. It is also possible to detect in-text evaluative segments and different aspects of the entities to which such segments refer, a task known as aspect-based sentiment analysis [33].

The traditional criticism to machine learning approaches (e.g. [34, 35]) is that they need to be trained on the same type of data that the classifier is going to be used with, that is, a classifier trained on a movie reviews dataset will not perform well on Amazon consumer product reviews. Current state of the art in sentiment classification is offered by unsupervised machine-learning approaches in the form of neural networks that use Transformers [31]. Language models based on the Transformer architecture have been shown to improve on previous top benchmark scores across numerous NLP tasks, both in natural language understanding and generation, including sentiment analysis [36]. The Transformer architecture, being unsupervised, might be thought to alleviate the need for domain-specific datasets; however, although pre-trained language models exist that can be used "off-the-shelf", these models still required to be fine-tuned on smaller, domain-specific, datasets using traditional supervised learning techniques.

Regardless of their accuracy, machine learning classifiers pose a second important limitation for our objectives, however, as they can help us assess what proportion of texts (or sentences) are positive or negative but tell us nothing about which words and expressions were identified that led to these classification results. A machine learning (or deep learning) sentiment classifier is a predictive model that acts as a black box which can provide no explanations as to how the classification result was arrived at, only (in certain cases) a confidence score. This is where lexicon-based sentiment analysis systems can be useful, since they determine the overall sentiment of a text by identifying specific sentiment words and phrases, which they can of course list in their output along with the classification.

These systems, however, do require rich lexical knowledge to achieve good results in different domains and their quality and coverage are the most determining factor in their performance. Thus, a high-quality lexicon is necessary to achieve good performance, along with a system that is able to process the linguistic context to account for the above-mentioned sentiment shifters, such as inversion (e.g., negation), intensification, and downtoning, in the form of context rules. Examples of this type of system include SO-CAL [37] and Lingmotif [38], a multilingual sentiment analysis tool that offers these features and whose English lexicon has been shown to be high-coverage and high-quality. In this work we employ the latter and we do not expect this tool to be 100% accurate, since aspects such as connotation, irony, sarcasm, discourse structure, etc. pose serious problems to automatic analyzers of this kind (see [39] for a thorough description of linguistic aspects that intervene in the expression of evaluation). Still,

we need to know what level of accuracy we have with our own domain-specific texts; thus, we carried out our own performance evaluation. We describe the tool and the experiment's results in section 3.3.

# 3. Materials and methods

## 3.1. Hypothesis

Our hypothesis is that there are significant short-term sentiment changes in economic terms once they become 'event words'. Thus, in this study, we aim to identify and track such sentiment shifts by analyzing the sentiment of the contexts in which those event words occur. We consider that terms become event words in the news when there is a sudden rise in their relative frequency. Conceptually, the task of discovering sentiment shifts from data can be formulated as follows: given a time-annotated corpus containing news items, the task is to discover changing sentiment trends in sentences containing 'inflation' in different time periods and tracing its collocational changes.

Fernández-Cruz [40] considers 'inflation' a neutral-sentiment economic marker, as it does not have an inherent sentiment, and its classification relies entirely on the expert's interpretation of inflation figures. Its semantic orientation can be described as a potentially negative indicator [41], as rising inflation figures (i.e., high inflation) are considered to have negative sentiment and vice versa. However, as discussed in the data analysis section, extremely low inflation figures are considered unfavorable.

## 3.2. Data sources

The data source is the Great Recession News Corpus (GRNC) [42], a specialized (economic/ financial domain) text compilation composed of approximately 35,000 articles from the business sections of *The Guardian* (52.9%) and *The New York Times* (47.1%) published between 2007 to 2015. These two newspapers are among the top influential newspapers worldwide [41] and they are considered to be ideologically in the center-left ideological spectrum. Divided into annual subcorpora, the latest version of the corpus contains approximately 26 million words. Finally, to enable ease of analysis and replication, filenames in the corpus are encoded in the format of [YYMM-Newspaper Code-File Number] and they are specified in the examples we provide in Section 4. All datasets and results are available in [43] to ensure replicability.

We obtained two datasets: the first one consists of all sentences (N = 5,481) containing the search term 'inflation' in the GRNC, along with a set of sentiment scores and other textual statistics generated by Lingmotif. Secondly, we extracted a dataset of the collocates of the term 'inflation' by year. This time interval was selected as it permits a concise yet detailed analysis and avoids seasonality. The LogDice statistical measure [44] was chosen with a cut-off value of 7, a window size of 5 words on either side, and requiring a minimum frequency of 5 instances for the collocate in the corpus and 3 for the collocate in the given range. Function words and proper nouns were removed, as they do not convey sentiment by definition. LogDice subsumes the frequency and exclusivity of collocations and returns a normalized measure in the range 0–14, which reduces the subcorpus size bias [45].

## 3.3. Instruments

Lingmotif [38], the sentiment analysis tool employed in this study, is a multilingual lexicon-based suite that can be used with general languages and domain-specific texts to calculate sentiment and textual values. It is implemented as a Python 3 library that is accessed through a user-friendly web interface using a REST API. By identifying linguistic expressions of polarity,

Lingmotif determines the semantic orientation of texts and sentences, and also provides rich quantitative information in the form of various sentiment and text metrics. The tool splits a text in one or more segments of the same size and returns partial and total sentiment metrics, along with a comprehensive set of text stats (e.g., number of nouns or the number of positive/negative words). Our study is based on Lingmotif's main sentiment metric, the Text Sentiment Score (TSS), a quantitative scale in the range of 0–100. It quantifies the sentiment of the analyzed text from extremely negative (0) to extremely positive (100). A value of 50 on the TSS scale is considered neutral. Importantly, Lingmotif arrives at a neutral score in one of two situations: either no sentiment words are found or the positive and negative words found cancel each other out. This implies that even scores slightly higher or lower than 50 are strong indicators of positive or negative polarity.

The application relies heavily on Lingmotif-lex's lexical resources [46], an English sentiment lexicon with over 26,000 single-word forms and over 41,000 multi-word items that have been manually curated to have wide coverage and be domain-neutral. Lingmotif allows the base lexicon to be overridden via plugin lexicons to address domain-specificity issues in sentiment calculation. For this study, we used SentiEcon [47], a fine-grained specialized sentiment lexicon in the financial domain. It contains 6,470 entries for both single and multiword expressions. A comprehensive set of sentiment shifters (nearly 1,000 in the current version of Lingmotif) was used to account for context-dependent valence modifications.

Even though both Lingmotif's core lexicon and the SentiEcon plug-in lexicon have been evaluated with good results in the past, it is necessary to ascertain whether TSS classification with this particular dataset is accurate enough. Thus, we carried out a formal evaluation study using a random sample (n = 360) from the full set of sentences (N = 5,481) in our main dataset (Confidence Level = 95%, Margin of Error = 5%). Three annotators independently labelled the sample dataset as belonging in one of three classes: positive (TSS>50), negative (TSS<50), neutral (TSS = 50). In order to avoid bias towards the lexicon-based classifier, annotators were asked to classify the sentences according to the overall sentiment they thought the sentences expressed, regardless of the number of sentiment words that were present in them. In order to ensure annotation reliability, Inter-Annotator Agreement (IAA) was calculated using the Disagree Python package [48]. Agreement was found to be very high for both Fleiss's kappa ($\kappa$ = 0.878) and Krippendorff's alpha ($\alpha$ = 0.878). The gold standard was arrived at by majority vote on differing labels, as no case was present where all three annotators disagreed on the label.

Precision and recall evaluation metrics were calculated using the scikit-learn [49] Python library. With an overall accuracy of 76% and a macro-averaged F1 score of 0.73, the classification predictions can be said to be very acceptable for a multi-label classification task. The lowest scores were found to be for the neutral class, as Lingmotif's neutral classification can be arrived at in either of two very different ways: when no sentiment expression is found or when the positive and negative expressions cancel each other out. Table 1 shows the classification with all relevant evaluation metrics.

**Table 1. Lingmotif's evaluation metrics for three-way sentiment classification.**

|  | Precision | Recall | F1-score | Support |
|---|---|---|---|---|
| NEG | 0.68 | 0.88 | 0.76 | 136 |
| NEU | 0.60 | 0.75 | 0.67 | 40 |
| POS | 0.92 | 0.67 | 0.77 | 184 |
| Accuracy |  |  | 0.76 | 360 |
| Macro avg | 0.73 | 0.76 | 0.73 | 360 |
| Weighted avg | 0.79 | 0.76 | 0,76 | 360 |

Finally, our study also utilizes Lancaster Stats Tools [50], an online statistical tool, and The Sketch Engine [51], a web-based tool that extracts linguistic information, (in our case, relative frequencies per million tokens, collocations and concordances) from large corpora. We uploaded the GRNC to The Sketch Engine in XML format so that the subcorpora feature was available. Relative frequency was calculated by dividing the number of occurrences of 'inflation' in each yearly subcorpus by the total number of words in the subcorpus, and then multiplying it by 1 million.

We also use Lancaster Stats Tool to carry out Usage Fluctuation Analysis (UFA). This method attempts to examine the fluctuation of word usage manifested through collocation, that is, the co-occurrence of words in texts. The instrument calculates Gwet's AC1 agreement statistic, a measure of inter-rater reliability that is used to assess the level of agreement between two or more raters when assigning categorical ratings to a set of items. McEnery et al. [52], the original proponents of this method, use this statistical measure to calculate the agreement between different lists of collocates in a timeline. The technique relies on two simple assumptions: (i) words co-occurring in the vicinity of other words provide insight into the words' usage (collocation principle), and (ii) the change in the pattern of co-occurrence of words over time can identify points where their usage changes. The tool returns a set of collocation lists: CONSISTENT for collocates that do not change in the timeline), INITIATING (collocates that emerge after the second data point and remain until the end of the series), TERMINAL (collocates that emerge in the last data points, and TRANSIENT (collocates that emerge after the second data point and decline before the final data points).

## 3.4. Method

Our procedure to examine discover and study semantic orientation changes of terms over time consists of four steps, which we describe below.

In step 1, we process the sentences dataset with Lingmotif and compare the yearly TSS means. To do so, we perform annual parametric (one-way ANOVA, post-hoc tests: Bonferroni adjusted t-test, p-values) and non-parametric [53] statistical analyses with the Lancaster Stats Tools online [50]. The hypotheses are as follows: (i) H0: $\mu1 = \mu2 = \mu3\ldots = \mu9$, and (ii) H1: means are not all equal.

In step 2, we observe general trends in a more detailed manner. To do so, we aggregate the text sentiment score (TSS) figures by quarterly averages and calculate the relative frequency of the terms for each of the 36 data points of the time series (corresponding to quarters). The dispersion points are fitted and analyzed using the peaks-and-troughs [54] technique in order to measure the difference in sentiment/frequency between consecutive points during the period. Peaks-and-troughs is a technique used in time series of diverse nature to identify trends in data by analyzing the high (peaks) and low (troughs) points of a chart over a certain period of time.

In step 3, we apply Usage Fluctuation Analysis to the dataset of collocations by year, which provides a concise and reliable comparison throughout a specific timeline. The parameters used in the tool were as follows: LogDice>7, Consistency Threshold 80%, 1 sampling point, Absolute Frequency cut-off = 0.

In step 4, a small number of representative collocations are selected for qualitative analysis by examining concordance lines to provide a more detailed picture of how the focus terms are characterized through the time series.

## 4. Analysis

### 4.1. Sentiment analysis

Table 2 provides descriptive statistics of several basic text metrics by year for the first target term. Fig 3 shows the distribution plot throughout the corpus. There was a statistically

**Table 2. Statistic summary for 'inflation'.**

| Year | N | Rel. Freq. (pmt) | TSS AVG | TSS StDev |
|---|---|---|---|---|
| 2007 | 154 | 64.84 | 51.21 | 34.40 |
| 2008 | 337 | 135.27 | 40.49 | 34.54 |
| 2009 | 147 | 62.30 | 46.19 | 33.52 |
| 2010 | 238 | 90.53 | 46.24 | 36.45 |
| 2011 | 633 | 193.29 | 42.61 | 33.15 |
| 2012 | 505 | 137.82 | 46.20 | 33.69 |
| 2013 | 671 | 147.00 | 50.82 | 35.27 |
| 2014 | 1100 | 222.13 | 50.51 | 34.12 |
| 2015 | 1696 | 358.33 | 48.68 | 34.81 |
| **Total** | **5481** | **175.43** | **47.77** | **24.57** |

significant effect of diachrony on TSS values (one-way ANOVA, $F_{(8, 5481)} = 5.75$, $p < 0.001$; Kruskal-Wallis $H_{(8)} = 47.79$, $p < 0.001$). The effect size was small ($\omega = 0.083$).

## 4.2. Peaks-and-troughs

Fig 4 plots the quarterly averages of TSS. To observe the trends more clearly, we processed these figures using a regression model (Linear GAM) of sentiment scores on a regression curve with 95% and 99% confidence intervals [55]. The form of the sentiment evolution of the term is that of a rotated s-shape. We identified two peaks, corresponding to 2007 and 2013–2014 and one trough (2008–2012).

In Fig 4, first we observe a steep increase in negative sentiment in 2007. The second phase (2008–2012) is characterized by a predominant negative sentiment and intermittent stabilization, with a discernible shift to a positive sentiment beginning to take shape around 2010. The last interval corresponds to the years 2014–2015, with a sharp rise to positive levels (TSS>50), followed by a fall to the threshold of negativity (TSS<50) in 2015. The term 'inflation' becomes an event word in times of change, as relative frequency in the corpus rises significantly in 2008, 2011, and 2014–2015. The stages are outlined as follows:

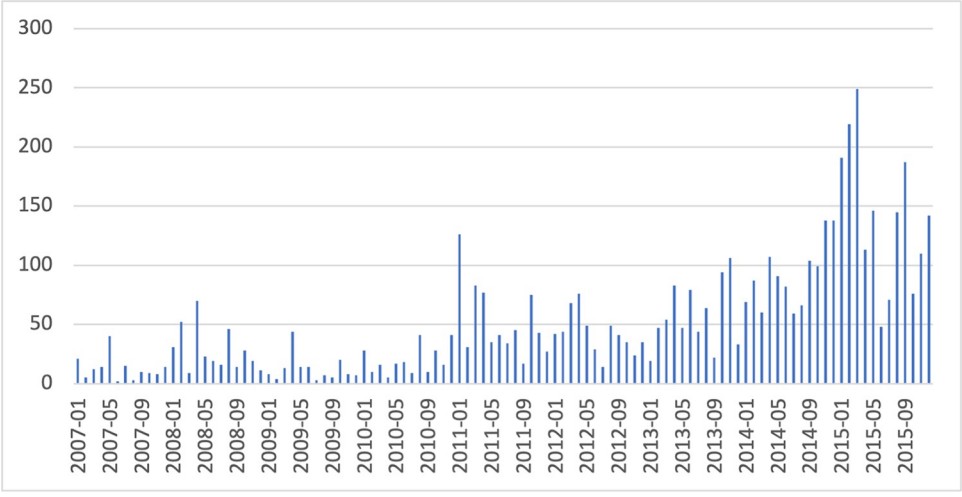

**Fig 3. Distribution plot of 'inflation' trough the corpus.**

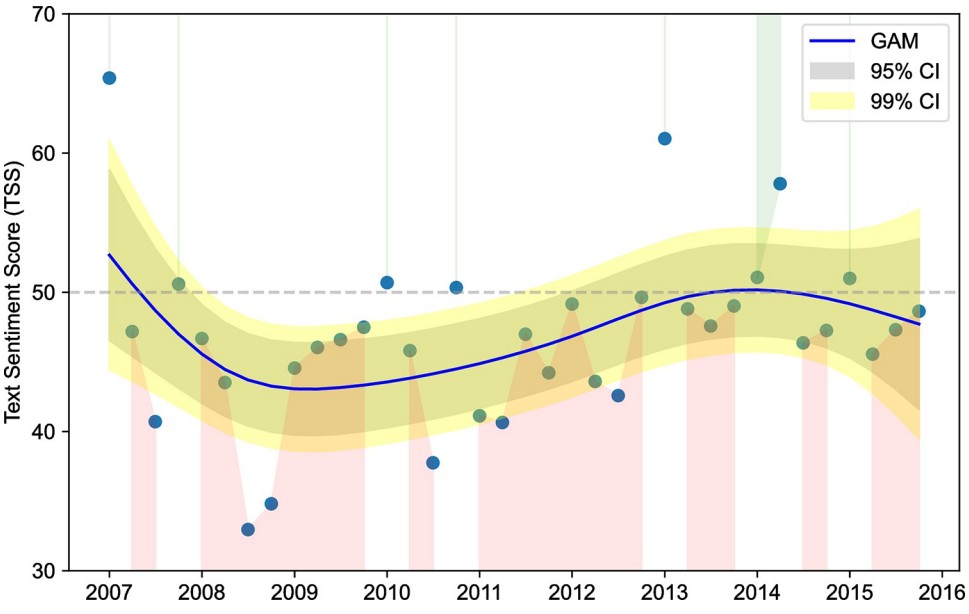

**Fig 4. Quarterly peaks-and-troughs for TSS.**

1. In the first period (64.84 occurrences per million tokens in the GRNC), the average TSS starts from a positive sentiment in 2007 (TSS = 51.21) so that, in relative terms, most sentences referring to inflation are positive in nature.

2. The second period (2008–2012) starts with a steep decline of 10.72% and TSS = 40.49 in 2008, after which scores stabilized, rising slightly to approximately 46 in 2009, 2010, and 2012. The fall in sentiment values is not accompanied by a sharp rise in the relative frequency of the term in the corpus; rather, it is irregular over this period (135.27 per million tokens in 2008, a peak of 193.29 in 2011).

3. The third period (2013–2015) encompasses the time after the policy changes in the European Central Bank. This reflects a pronounced upward trend in sentiment score averages, reaching TSS = 50.51 (i.e., slightly positive) in 2014, coinciding with Quantitative Easing policies, to subsequently fall again below the threshold of negativity in 2015 (TSS = 48.68), after the Eurozone's drop to negative inflation in the first quarter. After an initial period of continuity in the level of frequency in discourse, the frequency increased in the 2013–2015 interval (2013:147.00; 2014:222.13; 2015:358.33 per million tokens).

## 4.3. Usage fluctuation analysis

The next step is usage fluctuation analysis (UFA), which identifies 10 CONSISTENT collocates across the nine periods. Table 3 presents a comprehensive chart of the results summarized in Fig 5, indicating the sentiment categories of the most significant collocates though the timeline. The results show that the bulk of shifts in word use (TRANSIENT collocates) occurred after the outbreak of the 2008 crisis, when the term became an event word and negative collocational patterns emerged.

This emergence of new collocates indicate a period of pejoration in the sentences that include 'inflation' and it is consistent with TSS figures. The interval from 2011 onwards on both TRANSIENT and TERMINAL collocates signals semantic amelioration.

**Table 3. UFA analysis of the collocates of 'inflation'.** <u>Underlined</u> words convey positive sentiment, words in *italics* convey negative sentiment.

| | |
|---|---|
| CONSISTENT | down (2007–2015), expectation (2007–2015), increase (2007–2015), keep (2007–2015), level (2007–2015), measure (2007–2015), pay (2007–2015), pressure (2007–2015), record (2007–2015), remain (2007–2015), risk (2007–2015) |
| TERMINAL | bank (-2007), cent (-2007), offset (-2007), trade (-2007), control (-2013) |
| TRANSIENT | **2007**<br>combat (2007–2010), income (2007–2011), high (2007–2013), run (2007–2013)<br>**2008–2012**<br>*bad* (2008–2009), double-digit (2008–2010), account (2008–2011), commodity (2008–2011), curb (2008–2011), fight (2008–2011), *soar* (2008–2011), sharply (2008–2012), surge (2008–2012), currency (2008–2013), reason (2008–2013), sharp (2008–2013), *fear* (2008–2014), slow (2008–2014), annual (2008–2015), cause (2008–2015), central (2008–2015), *concern* (2008–2015), consumer (2008–2015), core (2008–2015), datum (2008–2015), december (2008–2015), economic (2008–2015), economist (2008–2015), economy (2008–2015), exclude (2008–2015), figure (2008–2015), food (2008–2015), further (2008–2015), headline (2008–2015), hit (2008–2015), household (2008–2015), index (2008–2015), interest (2008–2015), *negative* (2008–2015), pace (2008–2015), percent (2008–2015), push (2008–2015), raise (2008–2015), sign (2008–2015), start (2008–2015), *unemployment* (2008–2015), zero-m (2008–2015), prevent (2009–2011), august (2009–2014), may (2009–2014), control (2009–2015), cpi (2009–2015), drop (2009–2015), fall (2009–2015), outlook (2009–2015), real (2009–2015), target (2009–2015), accelerate (2010–2011), future (2010–2012), policy (2010–2012), finally (2010–2014), monetary (2010–2014), *threat* (2010–2014), average (2010–2015)<br>*hurt* (2011–2013), release (2011–2012), spending (2011–2012), *erode* (2011–2013), ease (2011–2014), lag (2011–2014), boost (2011–2015), continue (2011–2015), late (2011–2015), likely (2011–2015), mpc (2011–2015), persistently (2011–2015), rebound (2011–2015), show (2011–2015), temporary (2011–2015), report (2012–2014), slightly (2012–2014), back (2012–2015)<br>**2013–2015**<br>*fear* (2013–2014), outstrip (2013–2014), prolonged (2013–2014), bring (2013–2015), little (2013–2015), recent (2013–2015), <u>well</u> (2013–2015) |
| INITIATING | **2008–2012**<br>rise (2009-), basket (2011-), forecast (2011-), quarterly (2011-), rpi (2011-), <u>stable</u> (2011-), *weak* (2011-), eurozone (2012-), expect (2012-), percentage (2012-)<br>**2013–2015**<br>*deflation* (2013-), employment (2013-), period (2013-), <u>consistent</u> (2014-), currently (2014-), drive (2014-), gradually (2014-), low (2014-), outpace (2014-), pick (2014-), return (2014-), stay (2014-), term (2014-), again (2015-), <u>cheap</u> (2015-), close (2015-), <u>confident</u> (2015-), consistently (2015-), core (2015-), decline (2015-), downward (2015-), effect (2015-), even (2015-), explain (2015-), factor (2015-), february (2015-), forecast (2015-), further (2015-), <u>gain</u> (2015-), <u>help</u> (2015-), increase (2015-), largely (2015-), mean (2015-), month (2015-), move (2015-), near-term (2015-), objective (2015-), oil (2015-), pound (2015-), <u>reasonably</u> (2015-), return (2015-), see (2015-), *sluggish* (2015-), still (2015-), suggest (2015-), too (2015-), turn (2015-), underlie (2015-), very (2015-), *weakness* (2015-), weigh (2015-) |

CONSISTENT collocates all fall within the domain of economics, whether specialized (e.g., 'rate', 'growth') or directionality indicators ('low', 'fall', 'rise', etc.). While the sentiment of these items is generally neutral, some may be linked, even if not necessarily, to negative connotations ('pressure' or 'risk').

## 4.4. Qualitative analysis

**4.4.1. First stage (2007).** The yearly average score of 'inflation' in 2007 (TSS = 51.27) is positive. In general, these sentences refer to the daily evolution of economy before the advent of the crisis. Most of the TERMINAL UFA collocates belong in the domain of economics and end in 2007, as more prominent collocates will emerge in the following stage. Even if the averages for the second (TSS = 47.18) and third quarter (TSS = 40.71) are negative, all of the significant collocates for this stage belong to the domain of economy and none of them convey sentiment. For instance, its most significant collocation is 'adjust' (LogDice = 10.48), which is used to

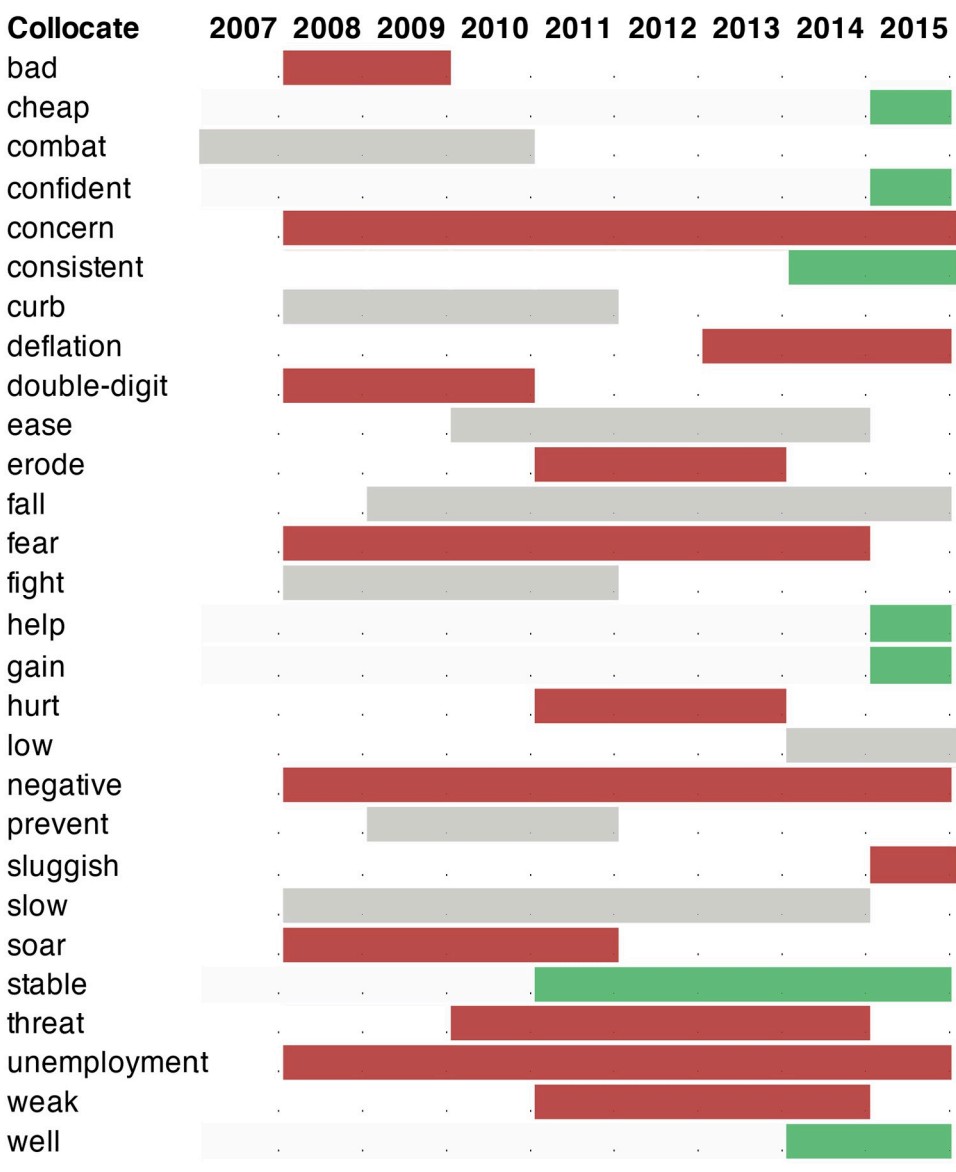

**Fig 5. A diachronic summary of relevant UFA collocates and their sentiment.** Red: negative, grey: neutral, green: positive.

create equivalences between two different historical prices. An example is shown in example (1) below.

(1) Yellowcake is trading at $90 a pound, nearing the record high, **adjusted** for inflation, of about $120 in the mid-1970s. 0703NYT140.xml

**4.4.2. Second stage (2008–2012).** The second stage of analysis (2008–2012) covers the extended time interval spanning the first five years after the Credit Crunch outburst. As inflation control was one of the workhorses of the central banks during this period, the term 'inflation' became an event word and received wide press coverage. The European Central Bank

and other central banks maintained inflation at artificially low levels, which reduced consumption and curbed the ability to create employment.

As can be observed in the TRANSIENT category of collocates, the context of the target term is characterized by more specific units (e.g., one of its indicators, the Consumer Price Index, 'CPI' (LogDice = 10.17 in 2009)) and emotion-related words such as 'threat' or 'fear' (Log-Dice = 8.21 and 7.98 in 2010, respectively), and verbs such as 'erode' (LogDice = 8.57 in 2011). Examples of such cases are (2) and (3).

The sentiment of fear instilled in the discourse of uncertainty provoked by uncontrolled levels of inflation, whether rising ('double-digit' LogDice = 8.96 in 2010) or decreasing ('negative', LogDice = 9.32 in 2009). The latter clearly alludes to the fear of negative inflation, one of the risks during this period, which is reflected in changes in sentiment and relative frequency. This occurs when the value of money increases and prices fall abruptly. Simultaneously, this is caused by an oversupply of products and a shortage of credit and money in circulation. As illustrated by examples (2) through (4), this atmosphere of uncertainty is reflected in the increased use of modality or conditionals, as well as the co-occurrence of lexical units with clearly negative sentiment (e.g., 'worsening' or 'contraction'). Although the consensus of central banks in Europe was to maintain low inflation levels at any cost, the press discourse also demonstrated the dangers that an eventual deflation could cause (5).

(2) They **fear** inflation **could whittle away** the value of cash holdings. 1001GU0019.xml

(3) In the developed economies, the **threat** of inflation **would** further stifle consumer confidence at a time when it is already extremely fragile. 1207GU00243.xml

(4) It is already a consensus view that core consumer inflation will turn negative soon, but we must watch if a **worsening** of the economy pushes Japan into a deflationary spiral, even though the Bank of Japan sees no signs of that happening right now. 0901GU0051.xml

(5) Experts are predicting a third consecutive quarter of **contraction** amid warnings that consumer price inflation risks slipping into the negative next year, while unemployment is approaching levels not seen since the dark days of the 1990s. 0812GU0012.xml

During 2011 and 2012, certain verb collocates signal rising concerns about increasing inflation levels following less restrictive policy changes. The most significant is the use of the verb 'erode' (LogDice = 8.57 in 2011) to illustrate the slow and progressive damage that inflation control and the lack of expansionary policies caused to different aspects of the economy. In fact, many of the concordance lines show that this word is used in news articles with a didactic function in a way that allows readers to understand the effects of such "bad" policies and clearly convey the message of corporate voices to the public. This is illustrated by examples 6–8.

The Eurozone reached a double-digit inflation rate (10.1%) in August 2022. During the 2008–2012 period, however, the shadow of high inflation was distant (3.35% in 2008 or 2.72% in 2011). The rhetorical use of the 'double digit' in the discourse to the detriment of other lexical selections, then, brought a negative connotation to the sentences and was thus perceived by the readers, who would then agree with the hegemonic discourse. As we can see in the examples 9–11, the use of the term in discourse is far from being just an objective description of events, as it alludes to the darkest times for the progressive public of *The Guardian*, with multiple mentions to the of the twilight of the Keynesian era, i.e., 'oil crises of the 1970s', 'Thatcherism', 'far cry' and the primary fear of hunger ('less food').

(6) The value of savings is **eroded**: If inflation out-paces interest rates, then it means the 'real' value of savings will fall. 1103GU0033.xml

(7) And Marc Ostwald of Monument Securities pointed out that the yield of just 1.93% means that people who bought the debt will see their returns comprehensively **eroded** by inflation. 1201GU00199.xml

(8) **Bad** policies might lead to high inflation, which **erodes** bond values, or a bond market might seize up at a time of financial stress. 1204NYT105.xml

(9) That's a far cry from the **double-digit inflation** rates that battered the economy at times in the 1970s, but still worrisome. 0802NYT.xml

(10) Thatcher at least had the excuse of fighting **double-digit inflation**. 1011GU0096.xml

(11) Economists said **double-digit inflation** was a cause for concern because it meant less food would be eaten by the poor in a country with an annual per capita income of only $835 but with pride in its having reduced poverty in the past decade. 0802NYT109.xml

**4.4.3. Third stage (2013–2015).**   In the third study period (2013–2015), yearly TSS averages rise for the first time to the positive threshold in 2013–14 (TSS = 50.82 and 50.51) and fall again to a negative average (TSS = 48.68) in 2015. In this period, 'inflation' co-occurrs with some of the infrequent positive UFA collocates (e.g., 'well' or 'help'). Also, in comparison with the 2008–2012 period, fewer negative UFA collocates emerge.

We consider that the connotative sentiment shift took place some time after the momentous *Whatever it takes* (also known as the *Draghi* moment) address by ECB President Mario Draghi in July 2012 (12). In his speech, he announced more progressive measures and promised a turnaround of the course of policies to save the European currency, which was severely threatened by the Greek crisis and the EU's self-imposed austerity measures. More than a decade on, this speech has become one of the milestones of the history of the European Union. Interestingly, "whatever it takes" became a very productive construction and was widely replicated in the news through the remaining timeline.

Downward directionality indicators ('low', and 'fall', LogDice = 10.01 and 9.30 in 2014, respectively) dominate the collocation rankings in 2014 and 2015 (see examples 13–15). In 3.1 we discussed that 'inflation' was considered a 'potentially negative' economic indicator. SentiEcon classifies the collocation of such 'potentially negative' words (e.g., 'inflation', 'interest rates', etc.) with descending directionality words (e.g., 'fall', 'drop') as positive (i.e., 'fall in inflation', 'interest rates dropped'). In our analysis of the combined GRNC 2013–2014 subcorpora on The Sketch Engine, we found 283 cases of 'inflation' co-occurring with the collocates 'low' and 'fall'. In order to ascertain their polarity, we analyzed a random sample of 20% (n = 56) of such sentences. Of these, 26 sentences (46.42%) were found to be positive, while 18 (32.14%) were negative and 12 (21.42%) neutral.

(12) President Mario Draghi has said the ECB will do **whatever it takes** to maintain inflation near a 2% target. 1501GU00213.xml

(13) Finally, **falling** inflation helps ease the squeeze on real incomes, increasing the chances that the recovery seen in recent months will persist. 1308GU00223.xml

(14) Along with the US and Germany, the UK economy is now expected to steam ahead as consumer spending rebounds, inflation remains **low** and unemployment continues to fall steadily. 1404GU00567.xml

(15) Danny Alexander, chief secretary to the Treasury, claimed the **lower** inflation figure as a victory for the Liberal Democrats. 1406GU00313.xml

In 2015, the TSS average (48.68) falls to the threshold of negativity. 'Low' is the main collocate of inflation (LogDice = 10.49) that year. In contrast with 2013–14, a superficial observation of sentences in the corpus that contained this collocate seemed to be used in a negative context. In order to ascertain our intuition, we extracted a random sample of 20% (n = 42) sentences from the 219 collocation occurrences in 2015. We found mostly negative sentences (24, 57.14%), while 6 were positive (14.28%) and 12 were neutral (28.57%). To illustrate this, examples 17–18 show that low prices can be good for consumption of the final product, and bad for macroeconomic figures related to debt.

We conclude the analysis with 2015's second most significant collocate: 'negative', mostly for 'negative inflation' (LogDice = 10.25) which took place while the Harmonized Index of Consumer Prices (a measure of inflation within the European Union) sunk below zero and can thus be considered the best candidate to understand why TSS figures return to negative levels. Examples 16–20 drawn from the 2015 subcorpus relate primarily to units with a prevalent negative sentiment, such as 'tumbling', 'stagnation', or 'fear'.

(16) "Given the **low** wage and price inflation data seen to date, and increased uncertainty about global growth, it will be particularly important for monetary-policy makers to closely monitor and depend on incoming data," he said. 1509NYT55.xml

(17) While such **low** inflation has been characterised by some government critics as a sign of economic fragility, for households it means they are better off in real terms. 1507GU00456.xml

(18) Mark Carney, the Bank's governor, has argued that while inflation is likely to turn **negative** during spring, Britain is not heading for the dangerous deflationary spiral feared in the eurozone. 1503GU000190.xml

(19) UK inflation forecast to remain **negative** Consumer prices index (CPI) has been dragged down by tumbling global commodity prices and the effects of a strong pound. 1511GU00185.xml

(20) Last week Piketty poured cold water on the Spanish government's claims that the crisis was over and insisted the eurozone still faced the risk of stagnation, weak growth and **negative** inflation. 1501GU00324.xml

## 5. Discussion and conclusions

We have examined how the institutionalized language use of news items changes in response to external events. The analysis and subsequent findings have provided a detailed perspective of how language usage can evolve quickly, over a limited time span, around one macroeconomic indicator during a significant economic crisis. Our approach, which combines the identification of sentiment and collocational shifts, avoids making wider assumptions or implications about broader language usage, instead focusing on the evolution of sentiment, semantic prosody, and collocations.

Quality newspapers' business sections offer samples of hegemonic discourse. These samples are rich in emotion expressions that "translate" experts' sentiment to influence public opinion.

Terms with high relative frequency become 'event words' in the press during crises, and they may undergo temporary, culturally motivated changes in sentiment, linked to the semantic phenomena of amelioration/pejoration that reflect in sentiment scores and collocations. Our results agree with the observations of [10], which evidence that this sentiment change has an indeterminate short-term duration (e.g., a few years). Usage Fluctuation Analysis and yearly

averaged sentiment scores provide significant quantitative and qualitative evidence of the postulate that semantic prosody is a non-static property of the lexicon, but variable over time. Our analysis shows that there were three clearly defined epochs during the timeline of the study: pre-crisis in 2007, the outburst of the crisis of 2008–2012, and the European debt crisis of 2013–2015. Usage Fluctuation Analysis provides further insights into this phenomenon, and evidence that the collocates of 'inflation' fluctuate and shift from being domain-specific during non-crisis time towards a compact system of sentiment collocates in times of crisis.

A sentiment perspective on the analysis of the context of economic indicators can provide a more nuanced understanding of their terminological nature. It allows to see beyond the traditional linear, one-dimensional definitions of terms and illustrate that they have a complex and fluid communicative potential. In addition, our results may contribute to the criticism of terminological synchronicity (for example, Temmerman [56] or Banks' [57] collective work on diachrony) as the usage of terms evolves even over brief time spans.

The final conclusion is discursive in nature. Quality press plays a key role as a mediator in the negotiation of term usage, and news items may influence reality. As such, understanding the interplay between linguistic structures and extralinguistic factors (i.e., macroeconomics, key economic actors, etc.) is essential in comprehending the complex nature of economic terminology and the broader economic context. According to Ferraro et al. [58], "the assumptions and ideas of economics come to create a world in which the ideas are true because through their effect on actions and decisions, they produce a world that corresponds to the assumptions and ideas themselves" (p. 12), which is in accord with the classic Thomas Theorem [59]: "If men [sic] define situations as real, they are real in their consequences" (p. 527).

Apart from the main insights of this paper, it is important to acknowledge its limitations and potential shortcomings. Our work covers only a rather limited ideological (yet influential) spectrum. It would be interesting to observe how the linguistic system (lexical, textual, etc.) of choices and omissions may shape their target public's perception of reality, which will be reproduced and transmitted to the speaker's community. Here, as a complement, a desideratum is an expansion of the GRNC so that a cross-ideological corpus study of the semantic evolution of terms can be achieved. Although it exceeds the scope of our paper, more comprehensive insights in our research could also be obtained by analyzing individual sentiment variation, such as differences between authors or newspaper articles and types.

As for future perspectives, the fluctuation of sentiment in terms is yet to be analyzed at different levels of specialization. Thus, it would be interesting to look above and below (i.e., professional discourse and general language). Future efforts may follow a similar approach to observe how the connotations of terms evolve in professional discourse, and their possible correlations with other levels of expertise.

## Supporting information

**S1 Data.**
(ZIP)

## Author Contributions

**Conceptualization:** Javier Fernández-Cruz, Antonio Moreno-Ortiz.

**Data curation:** Javier Fernández-Cruz.

**Formal analysis:** Javier Fernández-Cruz.

**Funding acquisition:** Antonio Moreno-Ortiz.

**Investigation:** Javier Fernández-Cruz, Antonio Moreno-Ortiz.

**Methodology:** Javier Fernández-Cruz, Antonio Moreno-Ortiz.

**Project administration:** Javier Fernández-Cruz, Antonio Moreno-Ortiz.

**Resources:** Javier Fernández-Cruz, Antonio Moreno-Ortiz.

**Software:** Javier Fernández-Cruz, Antonio Moreno-Ortiz.

**Supervision:** Javier Fernández-Cruz, Antonio Moreno-Ortiz.

**Validation:** Javier Fernández-Cruz, Antonio Moreno-Ortiz.

**Visualization:** Javier Fernández-Cruz.

**Writing – original draft:** Javier Fernández-Cruz.

**Writing – review & editing:** Javier Fernández-Cruz, Antonio Moreno-Ortiz.

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
