## [Decision Letter · Decision Letter 0]

27 Feb 2023

PONE-D-23-02829Tracking diachronic sentiment change of economic terms in times of crisis: connotative fluctuations of ‘inflation’ in the news discoursePLOS ONE

Dear Dr. Fernandez-Cruz,

Thank you for submitting your manuscript to PLOS ONE. After careful consideration, we feel that it has merit but does not fully meet PLOS ONE’s publication criteria as it currently stands. Therefore, we invite you to submit a revised version of the manuscript that addresses the points raised during the review process.

If you choose to revise the manuscript, please pay close attention to reviewers remarks (provided in a separate file). The review provides very clear insights into what should be revised in the manuscript. The main aspects are as follows:

<ul> <li> 

The corpus is not adequately described. In fact the reference [Fernández-Cruz et al., 2020] is mentioned only in the cover letter, but not in the manuscript and not in the bibliography section.

 This aspect must be revised.

 <li> 

Regarding the statistical analysis and methods: some aspects are described, but other aspects and some analyses are not properly described, especially for a wide audience (e.g. UFA, Gwet's A1)

 Also, it is not even clear what was measured . For example, line 300 says “we compare the yearly sentiment means.” What are they the means of? Per document? Per section? This only illustrates the need to provide a detailed account of what you measured and how you measured it (and not just mentioning the tool). Another example: table 1 - what is the relative frequency of? Of the term 'inflation' ? TSS and TSI AVG values – computed over what – documents? paragraphs? sentences? words?

 It would be also advisable to explain your terminology for a wider audience. For example (line 324) “reduces the subcorpus size bias” - what is it and why such reduction is important in the context of this research.

 Note that these are just examples, please make an effort to clarify and adequately explain all aspects in your Materials and Methods section.

 <li> 

The theoretical claims in the introduction and in the conclusion do not closely follow the actual measures and analyses as presented. Please refer to the detailed review for specific suggestions.

We look forward to receiving your revised manuscript.

Kind regards,

Michael Flor

Academic Editor

PLOS ONE

Reviewers' comments:

Reviewer's Responses to Questions

**Comments to the Author**

1. Is the manuscript technically sound, and do the data support the conclusions?

Reviewer #1: Partly

2. Has the statistical analysis been performed appropriately and rigorously? 

Reviewer #1: I Don't Know

3. Have the authors made all data underlying the findings in their manuscript fully available?

Reviewer #1: Yes

4. Is the manuscript presented in an intelligible fashion and written in standard English?

Reviewer #1: Yes

5. Review Comments to the Author

Reviewer #1: An interesting and well-written study with innovative methodology; however, I suggest some revisions are still necessary for maximum benefit of readers unfamiliar with the study and its underlying arguments.

Please see separate attached document for my suggestions

6. PLOS authors have the option to publish the peer review history of their article (what does this mean?). If published, this will include your full peer review and any attached files.

Reviewer #1: No

---

## [Author Response · Author response to Decision Letter 0]

21 Mar 2023

Dear sirs,

The authors would like to thank the reviewers for sharing their thoughtful comments and advice. We consider that the proposed changes and suggestions have greatly contributed to improving our work. 

The changes and corrections can be found in the "Tracked changes" version of the manuscript highlighted in red colour. The segments that we have removed have been kept in this version (in strikethrough) to facilitate comparison with the older version. 

In the "Response to reviewers" document, we offer a detailed response to each of the comments made by the reviewers.

Yours sincerely,

The authors

---

## [Decision Letter · Decision Letter 1]

23 May 2023

PONE-D-23-02829R1Tracking diachronic sentiment change of economic terms in times of crisis: connotative fluctuations of ‘inflation’ in the news discoursePLOS ONE

Dear Dr. Fernandez-Cruz,

Thank you for submitting your manuscript to PLOS ONE. After careful consideration, we feel that it has merit but does not fully meet PLOS ONE’s publication criteria as it currently stands. Therefore, we invite you to submit a revised version of the manuscript that addresses the points raised during the review process.

We look forward to receiving your revised manuscript.

Kind regards,

Michael Flor

Academic Editor

PLOS ONE

Journal Requirements:

Additional Editor Comments:

The manuscript (R1) requires just some minor revisions.

Pleas pay close attention to the reviewer comments, specifically in the file named re_review - Copy.docx

Reviewers' comments:

Reviewer's Responses to Questions

**Comments to the Author**

1. If the authors have adequately addressed your comments raised in a previous round of review and you feel that this manuscript is now acceptable for publication, you may indicate that here to bypass the “Comments to the Author” section, enter your conflict of interest statement in the “Confidential to Editor” section, and submit your "Accept" recommendation.

Reviewer #1: (No Response)

2. Is the manuscript technically sound, and do the data support the conclusions?

Reviewer #1: Yes

3. Has the statistical analysis been performed appropriately and rigorously? 

Reviewer #1: I Don't Know

4. Have the authors made all data underlying the findings in their manuscript fully available?

Reviewer #1: Yes

5. Is the manuscript presented in an intelligible fashion and written in standard English?

Reviewer #1: Yes

6. Review Comments to the Author

Reviewer #1: I just have some final suggestions for editing the manuscript before it is finalised and published. See separate file for these

7. PLOS authors have the option to publish the peer review history of their article (what does this mean?). If published, this will include your full peer review and any attached files.

Reviewer #1: No

---

## [Author Response · Author response to Decision Letter 1]

29 May 2023

Dear Dr. Flor,

Thank you for your prompt response regarding the submission of our manuscript PONE-D-23-02829. We appreciate the time and effort invested by the academic editor and reviewer(s) in evaluating our work. We have carefully reviewed the comments and suggestions provided, and we are grateful for the opportunity to revise and resubmit the manuscript.

Kind regards,

The authors

---

## [Editor Report · Decision Letter 2]

2 Jun 2023

PONE-D-23-02829R2Tracking diachronic sentiment change of economic terms in times of crisis: connotative fluctuations of ‘inflation’ in the news discoursePLOS ONE

Dear Dr. Fernandez-Cruz,

Thank you for submitting your revised manuscript to PLOS ONE. After careful consideration, we feel that it has merit but does not fully meet PLOS ONE’s publication criteria as it currently stands. Therefore, we invite you to submit a revised version of the manuscript that addresses the points raised during the review process. Version R2 has merit but does not fully meet PLOS ONE’s publication criteria as it currently stands. Therefore, we invite you to submit a revised version of the manuscript that addresses the points outlined below.

We look forward to receiving your revised manuscript.

**I am aware that you have requested expedited processing  ** 

Kind regards,

Michael Flor

Academic Editor

PLOS ONE

Journal Requirements:

**Additional Editor Comments:**

I have closely read the manuscript and I think revision R2 mostly addresses the reviewer recommendations made after revision R1.

There are still issues than need to be resolved, mostly technical, but not only.

Here is a list of points that need to be addressed:

1.

Line 99: "In contrast to the Federal Reserve" -- add "in the USA"

2.

Line 109: "that raised the existence" - what does 'raised' mean here? probably a different word is needed

3.

Line 155: "Waldron [11" -- should be "Waldron [18]"

4.

Line 161: "We adhere to Lyons’ [12," -- the reference should be "[20,"

5.

Line 167: "[l]anguages.." -- make it "languages..."

6.

Line 199: "Louw [19, p. 157] defined semantic..." -- Louw reference is [26]

7.

Lines 208-219: except for a short survey of detractors, what is the purpose/message of this paragraph? It is unclear. In the review of R1, a reviewer has also mentioned about this "Basically, the point isn’t to include the references just for the sake of having mentioned them, but to engage with the underlying issue regarding how the authors conceptualise/approach this in their own work."

The current paragraph in R2 is just mentioning detractors of semantic prosody.

The statement on lines 217-219: "The paper emphasizes the need to differentiate between collocational patterning and the connotations associated with a lexical item"

is not clear. Where is this differentiation reflected?

Maybe omit this whole paragraph.

8.

Line 221-222: "and progressively over time collocates more and more frequently" -- the sentence seems broken around the word 'collocates'. Rewrite this sentence.

9.

Line 240: "topics, and their attributes” [22, p. 259]." -- reference number [22] is not correct. Maybe 36?

10.

Lines 250-251: "(specifically, Transformers-based models, such as BERT) offer current state-of-the-art results for SA tasks [38]" --

Reference 38 is about machine translation. For sentiment, try a different paper,

maybe https://aclanthology.org/N19-1035/

11.

Line 260: "The traditional criticism to machine learning approaches (e.g., [30])" -- wrong reference [30].

12.

Note: Publication [41] (Taboada) is in the bibliographic list, but is not referenced in the manuscript text.

Consider mentioning it in section 2.3

13.

Line 303: "is a sudden rise in relative their frequency"  "their relative frequency"

14.

Lines 329-330: "We opted for the LogDice statistical measure (14-LogDice(7), L5-R5,C5,NC35),"

a) Please provide a reference to the LogDice original paper

Rychly, P. (2008). A lexicographer-friendly association score. In P. Sojka & A. Horak

(Eds.), Proceedings of recent advances in Slavonic natural language processing,

RASLAN (pp. 6–9). Brno: Masaryk University.

b) Please explain "L5-R5,C5,NC35" - those are not self-evident.

15.

lines 326-327: "For our research purposes, individual variation (e.g., authors, newspapers) was relevant."

It is not clear how relevant it was, i.e. where is it reflected in the results?

Maybe omit this sentence.

16.

Lines 373-380 (evaluation study):

Manual annotations are into 3 classes (positive, negative, neutral),

but TSS is on the scale 0-100.

Please describe how do you convert TSS values into (positive, negative, neutral)

(using cutoffs ?), so as to get 3-class results that are presented in Table 1.

17.

Lines 383-385: "Sketch Engine [56], a web-based tool that extracts linguistic information, (in our case,

relative frequencies per million tokens, collocations and concordances) from large corpora."

It is not quite clear how you used the Sketch Engine. Did you upload the GRNC to the SketchEngine?

Did you calculate frequencies over whole GRNC, or just the 'inflation' subset, or in other corpora? State it explicitly.

18.

Section 3.4 "Method".

This section lists four "steps" of analysis (those are separate analyses),

but it does not provide the results.

The results are listed in section 4.

This arrangement makes it difficult for the reader to follow the logic/description of analyses and their results (due to separation in the manuscript).

Please consider rearranging the text so that each description of analysis is followed by its results,

and then switch to the next set of analyses.

19.

Line 413: "analyzed using the peaks-and-troughs [59] technique"

Reference 59 is Gabrielatos&Marchi2012, but that presentation is about keyness, it is not about the peaks-and-troughs method

( peaks-and-troughs are not even mentioned in those slides).

Please cite a proper source.

20.

Figures 4 and 5: the figures are presented with clipped range on the Y axes.

Please consider adding a version the Fig.4 chart with the full range of the Y-axis, so readers can

get a fuller impression of the fluctuations relative to the full 0-100 range of the scale.

The values on Fig.4 seem to be in the narrow range 40-51, which is very close to the general 'neutral' line of 50.

Or maybe mark on the chart which zones of the Y-axis are negative/neutral/positive

(maybe use the cutoff values from the evaluation study?)

21.

Lines 444-463:

"In Fig 4, we identify the first stage in 2007, which corresponds to the stage prior

to the outbreak of the crisis, and a second stage of six years that starts with a pronounced

decline in negative sentiment values in 2008 and remains slightly upward until 2013."

This description does not seem to be corresponding to what I see in Figure 4.

For 2007 the line goes steeply down, which indicates growing negative sentiment.

The growth of negative sentiment slows in 2008. You use the word 'decline'

This is not adequate for two reasons. First, it is incongruent with the graphical depiction - in the chart going-down (or declining) means become more negative,

whereas you mean that negativity trend lessens (so decline in negativity is line-going-up-in-the-chart).

Negativity seems to be stable in 2009 and the change to positive seems to begin in 2010.

For the whole section of lines 444-463, the range of obtained TSS values is very close to 50.

Is 40, and 46-47, really negative or is it still within the lower bounds of 'neutral'?

is 51.21 really positive? or just firmly within neutral zone? The claim that 51.21 is 'positive' seems dubious.

Please state what are the negative/neutral/positive zones for the 0-100 TSS scale.

22.

Figure 6.

Please explain what do the colors red/green/grey mean in figure 6.

The manuscript text seems to disregard the coloring.

What did you try to convey there?

23.

Lines 485-486: "The term ‘inflation’ in the 2007 sub-corpus occurs in contexts with mainly positive connotations".

How this claim can be reconciled with Fig.4 which shows rapid growth towards more negative sentiment in 2007?

So the overall sentiment was declining (becoming negative) but somehow 'inflation' managed to sit in positive contexts?

This is very strange.

The TSS was computed only over 'inflation' sentences from GRNC, right?

24.

a)

Lines 563-564: "In the third study segment (2013-2015), ‘inflation’ co-occurrs with more positive sentiment words. Specifically, directionality indicators (‘low’, and ‘fall’,..."

Problem here is that neither 'low' nor 'fall' are positive sentiment words.

'Fall' (in the sense of falling) might even have some generic negative connotation.

So how do you derive that statement?

b)

Lines 567-568: "A cursory glance at the 2013-2015 concordance lines clearly indicates positive results".

Please don't send readers to concordance lines (even if the data is made available). It is also very subjective and vague.

You must support your claims in the body of the manuscript! Use statistics when needed.

25.

Line 588: "In 2015 TSS scores fall to the threshold of negativity."

What is that threshold? Is it strictly "50"? Where is the 'neutral' zone?

26.

Line 595. Please explain what are "HICP figures".

27.

Almost all the description is section 4 is focused on TSS.

The TSI results are mentioned but mostly ignored.

This creates a startling imbalance.

Please consider saying something about the TSI results.

or maybe remove TSI from this manuscript, consider it for another publication?

---

## [Author Response · Author response to Decision Letter 2]

7 Jun 2023

Dear reviewers,

The authors would like to thank the reviewers and the Editor in Chief for their prompt response and sharing their comments and advice. We consider that the proposed changes and suggestions have greatly contributed to improving our work.

We ensured that all modifications in the citations have been referenced in the Rebuttal letter.

The changes and corrections can be found in the updated version of the manuscript highlighted in red colour. 

We removed the original Fig 5. Former Fig 6 is now Fig 5.

The segments that we have removed have been kept in this version (in strikethrough) to facilitate comparison with the older version. In the Response to reviewers document, we offer a point-by-point response to each of the comments made by the reviewers.

Kind regards.

---

## [Editor Report · Decision Letter 3]

13 Jun 2023

Tracking diachronic sentiment change of economic terms in times of crisis: connotative fluctuations of ‘inflation’ in the news discourse

PONE-D-23-02829R3

Dear Dr. Fernandez-Cruz,

We’re pleased to inform you that your manuscript has been judged scientifically suitable for publication and will be formally accepted for publication once it meets all outstanding technical requirements.

Kind regards,

Michael Flor

Academic Editor

PLOS ONE

---

## [Editor Report · Acceptance letter]

26 Jun 2023

PONE-D-23-02829R3 

Tracking diachronic sentiment change of economic terms in times of crisis: connotative fluctuations of ‘inflation’ in the news discourse 

Dear Dr. Fernandez-Cruz:

I'm pleased to inform you that your manuscript has been deemed suitable for publication in PLOS ONE. Congratulations! Your manuscript is now with our production department. 

Kind regards, 

on behalf of

Dr. Michael Flor 

Academic Editor

PLOS ONE